# Tumor-Colonizing *E. coli* Expressing Both Collagenase and Hyaluronidase Enhances Therapeutic Efficacy of Gemcitabine in Pancreatic Cancer Models

**DOI:** 10.3390/biom14111458

**Published:** 2024-11-17

**Authors:** Lara C. Avsharian, Suvithanandhini Loganathan, Nancy D. Ebelt, Azadeh F. Shalamzari, Itzel Rodarte Muñoz, Edwin R. Manuel

**Affiliations:** 1Department of Immuno-Oncology, Beckman Research Institute of the City of Hope, Duarte, CA 91010, USA; lavsharian@coh.org (L.C.A.); sloganathan@coh.org (S.L.); nebelt@coh.org (N.D.E.);; 2Irell and Manella Graduate School of Biological Sciences, City of Hope, Duarte, CA 91010, USA; ashalamzari@coh.org

**Keywords:** pancreatic ductal adenocarcinoma, fibrosis, desmoplasia, resistance, hyaluronan, collagen, bacterial vector, hyaluronidase, collagenase

## Abstract

Desmoplasia is a hallmark feature of pancreatic ductal adenocarcinoma (PDAC) that contributes significantly to treatment resistance. Approaches to enhance drug delivery into fibrotic PDAC tumors continue to be an important unmet need. In this study, we have engineered a tumor-colonizing *E. coli*-based agent that expresses both collagenase and hyaluronidase as a strategy to reduce desmoplasia and enhance the intratumoral perfusion of anticancer agents. Overall, we observed that the tandem expression of both these enzymes by tumor-colonizing *E. coli* resulted in the reduced presence of intratumoral collagen and hyaluronan, which likely contributed to the enhanced chemotherapeutic efficacy observed when used in combination. These results highlight the importance of combination treatments involving the depletion of desmoplastic components in PDAC before or during treatment.

## 1. Introduction

Pancreatic ductal adenocarcinoma (PDAC) is the third leading cause of cancer-related death and is expected to rise by 2030. At the time of diagnosis, 80% of patients have advanced disease that is ineligible for surgical resection, and 50% have metastasized tumors, leaving PDAC patients with an overall survival time of less than a year [1,2,3]. FOLFIRONOX or gemcitabine plus albumin-bound paclitaxel is the standard first-line treatment for PDAC patients, with a majority progressing to second-line treatments due to chemoresistance [3,4]. The extracellular matrix (ECM) in the PDAC, characterized by desmoplasia due to the excess production of collagen and hyaluronan (HA), is considered a major contributor to therapeutic resistance. Increased tissue rigidity, interstitial fluidic pressure, and a dampened immune response are outcomes of the highly dense tumor stroma. Previous studies have shown that drug penetration is significantly hindered with increased collagen and HA, and depletion of ECM components leads to better drug perfusion and decreased tumor growth in PDAC models [5,6]. In addition to serving as a biophysical barrier, the ECM components act as signaling molecules that influence immune responses within tumors. Excess collagen is linked to low CD8^+^ T cell infiltration and has been proposed as a biomarker for immunotherapy resistance, while the upregulation of HA production in tumor cells correlates with immunosuppressive macrophage phenotypes [7,8].

Collagen I is highly abundant in desmoplastic tumors and is a fibril-forming collagen that is crosslinked by LOXL2, a lysyl oxidase that is upregulated in the PDAC. Excessive collagen crosslinking leads to fibrosis and a stiff ECM that enhances the migration of tumor cells and promotes stem cell transformation [9]. PDAC cell metabolism and proliferation are reliant on collagen I. For example, proline availability is regulated by collagen biosynthesis, and signaling through the DDR1-NFκB-p62-NRF2 cascade leads to changes in mitochondrial protein expression [10,11,12]. Furthermore, PDAC cells use collagen I in an autocrine signaling loop to promote an immunosuppressive tumor microenvironment (TME) with limited T cell infiltration [13]. Collagen IV is also abundant in the PDAC and has been proposed as a serum biomarker, as tumor cells actively secrete collagen IV in both primary and metastatic tumors, forming irregular basement membrane-like structures [14]. Type IV collagen in the PDAC has been associated with metastasis and angiogenesis, and the production of collagen IV allows tumor cells to secrete and anchor to their basement membrane through integrin receptors, supporting tumor cell survival [15]. Collagen is deposited within the TME primarily by pancreatic stellate cells (PSCs) and cancer-associated fibroblasts (CAFs), and metastasis is often seen in murine models engrafted with PSCs or CAFs known to produce more collagen through their interaction with integrins on the tumor cell surfaces [16,17]. In addition to collagens, HA overexpression is prevalent in a majority of PDAC patients, as well as a plethora of other cancers, and is secreted from both tumor cells and stromal cells [17,18]. Low molecular weight HA expressed by PDAC cells interacts with cell surface signaling molecules to increase proliferation through the TLR4 signaling cascade and, in turn, increases lymphatic permeability, cell adhesion, and matrix remodeling to facilitate metastasis, local inflammation, and fibrosis [19,20]. Collagen I, collagen IV, and HA are abundant in PDAC stroma and are negatively correlated with patient survival and chemotherapeutic response [21,22], making these ECM proteins enticing therapeutic targets that may delay disease progression and increase susceptibility to chemo- and immunotherapy.

Collagenases and hyaluronidases, enzymes that degrade collagen and HA, respectively, are being explored in preclinical and clinical studies to sensitize PDAC tumors to the immune system and increase their permeability for therapeutic intervention [23]. Bacterial collagenases were first studied and isolated from *Clostridium histolyticum* due to their ability to degrade human tissue [24,25], while bacterial hyaluronidases were characterized by Karl Meyer in the 1970s for their pathological role [26]. The FDA has approved the use of bacterial collagenases and hyaluronidases for various diseases, including the localized treatment of solid tumors. Since collagen and HA play an important role in tissue integrity throughout the body, systemic administration is not a viable option, and the use of intratumoral injection for localized treatment remains challenging [27,28,29]. To overcome these challenges, we previously showed that attenuated *Salmonella typhimurium* engineered to express collagenase or hyaluronidase selectively colonize PDAC tumors in mice and degrade intratumoral collagen or HA, respectively, without off-target degradation in healthy tissues, such as the skin. Intratumoral ECM depletion by these agents led to increased sensitivity to immuno- and chemotherapy [30,31].

To attain a synergistic effect with increased ECM degradation and decreased tumor progression, we have employed ColH, a collagenase isolated from *Clostridium histolyticum* [32], and HylB, a hyaluronidase found in *Streptococcus agalactiae* [33], in order to effectively degrade the ECM within solid PDAC tumors. We have engineered the *E. coli* strain BL21 to express both ColH and HylB as a single inducible agent with tumor specificity (BL21-TAN). BL21-TAN efficiently colonizes tumors, expresses both enzymes, and depletes collagen and HA content in PDAC tumor models, leading to an enhanced therapeutic outcome when combined with chemotherapeutic treatments. We predict that BL21-TAN has the potential to serve as a combination treatment to increase the sensitivity of PDAC tumors to both chemotherapy and immunotherapy, thereby significantly improving the outcomes of patients diagnosed with PDAC.

## 2. Materials and Methods

### 2.1. Animals and Cell Lines

C57BL/6 and NOD/SCID gamma (NSG) mice were bred and housed at the City of Hope (COH) Biomedical Research Center (BRC). For all studies, the animals were handled according to standard IACUC guidelines under an approved protocol. The KPC cells were generously provided by Dr. Laleh Melstrom. The UPN cell line was generated at COH from de-identified patient PDAC resections. BxPC3 was obtained commercially from ATCC, Manassas, VA, USA (CRL-1687). The KPC and UPN cells were maintained in DMEM and BxPC3 in RPMI. Both media contained 10% FBS, 2 mM L-glutamine, 100 units/mL penicillin, and 100 μg/mL streptomycin. Prior to tumor cell engraftment, cells were passaged ≤ five times at ≤80% confluency.

### 2.2. E. coli Strains and Generation of BL21-TAN

The BL21 (DE3) chemically competent *E. coli* was obtained from New England Biolabs, Ipswich, MA, USA and cultured in LB media. The ColH (Genbank Accession no. BAA34542) and HylB (Genbank Accession no. AAA56749.1) amino acid sequences were used to synthesize a codon-optimized cDNA sequence inserted into the NcoI site of the pBAD-His-A bacterial expression vector (Biomatik, Kitchener, ON, Canada), with a c-Myc tag fused to ColH and a hexahistadine tag fused to HylB. The in-frame insertion of TAN into the pBAD vector allowed for the expression of ColH and HylB upon induction with L-arabinose. Plasmids were transformed into BL21 (DE3), spread onto LB plates containing 100 μg/mL ampicillin, and incubated overnight at 37 °C. Positive clones were identified by colony PCR.

### 2.3. Bacterial Growth, Viability, and Analysis of TAN Expression

The BL21-TAN was cultured in media with (induced, I) or without (uninduced, U) 0.1% L-arabinose at 37 °C and 225 rpm, for intervals ranging from 2 to 24 h. ColH was detected in bacterial lysates by Western blot analysis using a primary monoclonal mouse anti-Myc antibody (R&D Systems, Minneapolis, MN, USA). HylB was detected in bacterial lysates by Western blot analysis using a primary monoclonal goat anti-His antibody (R&D Systems, Minneapolis, MN, USA). The growth kinetics were monitored through absorbance readings at 600 nm (Genesys 30, Thermo Fisher Scientific, Waltham, MA, USA) for up to 24 h. For immunofluorescence, the uninduced and induced BL21-TAN grown for ~4 h were fixed with 4% paraformaldehyde at room temperature (RT) for 30 min, permeabilized with 0.1% Triton-X 100/PBS at pH = 7.2 and RT for 30 min, followed by lysozyme (Sigma-Aldrich, St. Louis, MO, USA, 100 μg/mL final concentration in 5 mM EDTA) treatment at RT for 45 min. The fixed/permeabilized bacteria were incubated with the primary antibody (1:100) for 30 min with shaking in a humidified 37 °C incubator, followed by incubation with a FITC-conjugated anti-mouse secondary (1:200, Abcam, Cambridge, United Kingdom) and DAPI for 30 min with shaking in a humidified 37 °C incubator.

### 2.4. Fluorometric Substrate Assays

BL21-TAN was cultured under uninduced or induced conditions for 4 h at 37 °C. To measure collagenase activity, substrates consisting of bovine skin collagen type I, human placenta collagen type IV, and pig skin gelatin conjugated to FITC (Thermo Fisher Scientific) were used. The reaction was started by the addition of BL21-TAN under uninduced or induced conditions. Hyaluronidase activity was measured using hyaluronan conjugated to FITC (Thermo Fisher Scientific). The fluorescence intensity was captured using an iBright FL1500 Imaging System (Thermo Fisher Scientific) and quantified in relation to the BL21-TAN uninduced control condition using ImageJ v.2.14.0/1.54f (NIH, Bethesda, MD, USA).

### 2.5. Immunohistochemistry/Immunofluorescence (IHC/IF)

FFPE tumors from C57BL/6 or NSG mice were sectioned (5 μm), transferred to glass slides, deparaffinized, and rehydrated. To determine the in vitro enzymatic function, slides were treated with BL21-TAN uninduced or induced for 4 h. To examine the in vivo colonization, expression, and function, the mice were treated with BL21-TAN or the BL21-eGFP control prior to tumor collection. Slides were stained with trichrome and coverslipped. The hematoxylin and methyl blue channels were separated using color deconvolution and thresholds were set to cover the positive staining area for hematoxylin (red), and the raw integrated density was used to measure the density of collagen. To determine the HA density, slides were stained with HA-binding protein (HABP). (Millipore Sigma, Burlington, MA, USA) BL21 colonization was detected using a primary anti-BL21 lysate antibody (R&D), ColH was detected using a primary monoclonal mouse anti-Myc antibody (R&D), and HylB was detected using a primary monoclonal goat anti-His antibody (R&D). Brightfield and fluorescent images were acquired with a 63× 1.4 NA Plan-Apochromat objective (630× total magnification). Stitching was performed using ZEN 2.3 Pro (Blue) software (Carl Zeiss Inc., Oberkochen, Germany). The quantification of brightfield images was performed using ImageJ (NIH) and the quantification of fluorescence intensity was performed using Quantitative Pathology & Bioimage Analysis (QuPath) software (v0.2.1, University of Edinburgh, Edinburgh, UK) [34].

### 2.6. Tumor Implantation, Administration, and Induction of BL21-TAN

For all experiments, 6–8 week-old mice were used. Either 2 × 10^6^ BxPC3 cells or 5 × 10^5^ KPC cells were subcutaneously injected into the right flank of the NSG mice or C57Bl/6 mice, respectively, in a volume of 100 μL PBS using a 27-gauge needle. Mice with palpable tumors (>150 mm^3^) were intravenously injected with 5 × 10^7^ colony-forming units (CFUs) of the BL21-TAN or BL21-eGFP control per day for three consecutive days. Forty-eight hours following the final BL21 administration, protein expression was induced by the administration of 40 mg L-arabinose through an intraperitoneal route. Forty-eight hours following induction, tumors were collected and analyzed via IHC/IF staining for colonization, enzymatic expression, and enzymatic function.

### 2.7. Combination Treatment Studies with Gemcitabine

Following the BL21 induction treatment, as described above, the mice were subsequently administered gemcitabine (15 mg/kg) or PBS 48 h post induction. Maintenance doses of gemcitabine or the PBS control were administered every three days thereafter.

### 2.8. Statistics

All statistical analyses were performed using the Prism software by GraphPad (v9). Data was analyzed by performing Student’s *t*-test and 2-way ANOVA. Unless otherwise indicated, all error bars represent the standard error of the mean.

## 3. Results

### 3.1. Construction and Inducible Expression of BL21-TAN

To obtain the inducible expression of ColH and HylB, we utilized the pBAD expression system, which employs the araBAD operon that is tightly regulated by L-arabinose availability [35]. The use of this system allows for the controlled expression of ColH and HylB within the tumor, limiting off-target toxicities that are expected with systemic collagenase and hyaluronidase expression [36,37,38]. We designed a construct that encodes for ColH with a C-terminal Myc tag and HylB with a C-terminal His tag under the regulation of the pBAD promoter. Each gene was preceded by an independent N-terminal ribosomal binding site to ensure the expression of two individual proteins upon induction (Figure 1A). The constructed plasmid was then transformed into BL21 *E. coli*, which was selected for its ability to colonize solid tumors and its efficiency in the production of recombinant proteins [39,40,41]. Colonies that integrated the plasmid were grown in media to the exponential phase (OD_600_, ~0.5) and placed under uninduced (U; 0% L-arabinose) or induced (I; 0.02% L-arabinose) conditions for four additional hours. Western blot analysis of the cell lysates and conditioned culture media (Figure 1B) confirmed the expression of both ColH and HylB under induced conditions, detected by their C-terminal tags. Minimal to no expression was observed in the uninduced samples, demonstrating a tight regulation of protein expression. The presence of both ColH and HylB in the culture media further suggests that these proteins are secreted by BL21 *E. coli* post induction. To determine the effect of TAN expression on bacterial growth kinetics, BL21-TAN was grown in uninduced or induced conditions alongside BL21-eGFP as an additional control, and the optical density (OD_600_) was measured at multiple time points over 24 h (Figure 1C). There was an observable initial increase in OD_600_ that stabilized by 16 h at an approximate OD_600_ of 5.0 in all three conditions, suggesting that ColH and HylB expression, overall, had no detrimental effect on BL21 growth. While we have demonstrated a tight regulation of protein expression and sufficient growth of bacterial cells under induced conditions, it is important that both ColH and HylB are expressed simultaneously in a single bacterium for further research. To observe the proteins, we performed immunofluorescent staining of the C-terminal Myc-tag and His-tag fused to ColH and HylB, respectively (Figure 1D). Staining of both proteins in a single bacterium was observed in cultures grown under induced conditions and was observed outside of the genomic DNA staining (DAPI). This confirms that both ColH and HylB are expressed simultaneously in the same bacterium and, in this case, are likely to be localized in the periplasm on their way to being secreted.

### 3.2. BL21-TAN Degrades Collagen I, Collagen IV, and Hyaluronic Acid In Vitro

To assess the enzymatic activity of ColH and HylB produced by BL21-TAN, we grew the engineered bacterial cultures under uninduced and induced conditions and performed in vitro functional assays utilizing FITC-labeled pig skin gelatin (Figure 2A), bovine skin collagen type I (Figure 2B), human placenta collagen type IV (Figure 2C), or HA (Figure 2D) as substrates. Following the co-incubation of FITC-labeled substrate with BL21-TAN, the extent of substrate degradation by enzymatic hydrolysis was measured by an increase in fluorescent molecules. Collagen type I and IV, two prevalent collagen types in the PDAC, were degraded when co-incubated with induced BL21-TAN, consistent with previous studies on the functional range of ColH [14,32]. Additionally, an increase in fluorescence intensity was observed in FITC-HA substrate solutions following the co-incubation with induced BL21-TAN, indicating the degradation of hyaluronic acid. These results confirm that ColH and HylB expressed by BL21-TAN are enzymatically active. Significant increases in fluorescence intensity between uninduced and induced conditions were observed in assays with FITC-collagen I (*p* < 0.0001, *t*-test), collagen IV (*p* < 0.0001, *t*-test), and HA (*p* < 0.0001, *t*-test), demonstrating that BL21-TAN is tightly regulated by the pBAD inducible expression system in vitro, with induction resulting in increased functional enzyme production. In contrast, no difference was observed between the uninduced and induced cultures co-incubated with FITC-gelatin, suggesting no effect on baseline gelatin degradation. This finding is in agreement with previous studies confirming that ColH does not exhibit gelatinase activity [32].

### 3.3. BL21-TAN Effectively Depletes PDAC Tumor-Derived Collagen and Hyaluronic Acid

Collagen and HA derived from healthy tissue differ in structure and function in comparison to tumor-derived collagen and HA. Collagen in healthy tissue is flexible with varying diameters evident between different tissue types, while tumor-derived collagen is thick, stiff, and aligned, leading to dense, fibrotic tissue that promotes tumor cell growth and metastasis [42,43]. HA metabolism is aberrantly regulated in tumor tissue compared to healthy tissue, resulting in an increase in the abundance of low molecular weight HA in tumor tissue, which promotes cancer cell proliferation, angiogenesis, and metastasis [44]. To determine the efficacy of BL21-TAN in degrading tumor-derived collagen and HA, we obtained serial sections of tumors grown from the murine-derived KPC pancreatic cancer cell line and human-derived pancreatic BxPC3 cell line, as well as a de-identified patient (UPN)-derived PDAC tumor. The KPC tumor models recapitulate the increased interstitial fluid pressure, hypoxia, and hypovascularity seen in the human PDAC, whereas the BxPC3 models display extensive collagen remodeling and increased stiffness within the tumor tissue, which is a prognostic marker for therapeutic response in human PDAC, [6,22,45,46,47,48] and UPN was used to more closely represent the human PDAC tumor microenvironment. Serial sections were treated with uninduced or induced cultures of BL21-TAN in vitro overnight. Tumor sections were then subjected to staining with trichrome to detect collagen (Figure 3A) or biotin-labeled HA binding protein (HABP; Figure 3B). The collagen (blue) content in the trichrome-stained sections was quantified by deconvolution using ImageJ and the raw integrated density of collagen was measured (Figure 3C and Appendix A), while QuPath was used to quantify HA density, determined by the relative fluorescence intensity of HABP (Figure 3D). A significant decrease in both the collagen density (*p* = 0.0014 for KPC, *p* = 0.005 for BXPC3, *p* = 0.0002 for UPN, *t*-tests) and HA content (*p* = 0.0065 for KPC, *p* < 0.0001 for BXPC3, *p* = 0.0029 for UPN, *t*-tests) was observed in sections treated with induced cultures of BL21-TAN when compared to sections incubated with uninduced cultures in all three tumors. These results suggest that ColH and HylB produced by BL21-TAN effectively deplete tumor-derived collagen and HA and further support the ability to regulate enzymatic function through induced expression of TAN.

### 3.4. BL21-TAN Expresses Both ColH and HylB in PDAC Murine Models

We evaluated the ability of intravenously delivered BL21-TAN to colonize tumors in C57BL/6 mice with subcutaneous (s.c.) KPC tumors and NSG mice with s.c. BxPC3 tumors through immunofluorescence analysis. BL21-eGFP was used as a vector control to display any observed effects caused by bacterial colonization alone. Following the intravenous administration of BL21-eGFP or BL21-TAN and the subsequent induction with intraperitoneal (i.p.) L-arabinose, BL21 staining was observed in both tumor models treated with BL21-eGFP and BL21-TAN, as expected (Figure 4). These results demonstrated effective tumor colonization by BL21, independent of the expressed genes. To ensure that BL21-eGFP or BL21-TAN were effectively clearing from the systemic circulation, we examined the spleens from treated mice, where bacteria from the bloodstream typically accumulate during infection [49]. Spleens from tumor-bearing mice treated with ampicillin-resistant BL21 were homogenized and subjected to microbial enumeration by serial plating on LB agar containing ampicillin. No bacterial colonies were detected, indicating that BL21 colonization was limited to the tumors. Together, these data strongly suggest that BL21 colonization is tumor-specific. Next, we assessed the in vivo enzyme expression following the systemic treatment of tumor-bearing mice with BL21-TAN. Antibodies against the Myc-tag on ColH and the His-tag on HylB were used to visualize the expression of bacterially produced enzymes after induction of the pBAD promoter. The KPC and BxPC3 tumors treated with BL21-TAN showed a dual expression of ColH and HylB, visualized by the similar staining pattern and yellow/orange color in the ColH/HylB overlaid panel (Figure 4), indicating the successful in vivo induction of the TAN construct and expression of both enzymes from a single bacterium. ColH and HylB expression was localized to regions with a high BL21 density, further confirming that BL21-TAN expresses both ECM-degrading enzymes within the tumor microenvironment in the two PDAC tumor models.

### 3.5. BL21-TAN Depletes Collagen and Hyaluronan in the Murine Model of PDAC

We demonstrated that BL21-TAN preferentially colonizes tumors and displays a dual expression of ColH and HylB; therefore, the next important question is whether these enzymes are functionally active under physiological conditions. To determine the functional capacity of the ECM-degrading enzymes produced by BL21-TAN, the BxPC3 tumors in NSG mice treated with either BL21-eGFP or BL21-TAN were analyzed for collagen (blue) and HA abundance using trichrome (Figure 5A) or HABP (Figure 5B). The BxPC3 tumors from mice treated with BL21-TAN exhibited significantly lower collagen levels compared to tumor sections from mice treated with BL21-eGFP (*p* < 0.0001, *t*-test; Figure 5C). Similarly, IHC staining of hyaluronan revealed significantly less hyaluronan in tumors from mice treated with BL21-TAN compared to tumors treated with BL21-eGFP (*p* < 0.0001, *t*-test; Figure 5C). These data suggest that BL21-TAN effectively degrades intratumoral collagen and HA under physiological conditions.

### 3.6. Pre-Treatment with BL21-TAN Increases Gemcitabine Efficacy in the Murine Model of PDAC

Desmoplasia in PDAC tumors has been shown to be a pathological barrier, hindering drug penetration by affecting vasculature, increasing interstitial pressure, and increasing organ stiffness, which leads to poor small-molecule perfusion [50,51]. Therefore, we hypothesized that PDAC tumors treated with ECM-degrading BL21-TAN would lead to an increased efficacy of chemotherapeutic interventions. The NSG mice bearing BxPC3 tumors were treated with BL21-eGFP or BL21-TAN for three consecutive days and induced with 40 mg L-arabinose delivered via an i.p. injection twenty-four hours after the final BL21 dose. Control groups receiving PBS only or gemcitabine only were administered PBS. Gemcitabine was administered by i.p. injection 48 h post induction and then every three days for a total of twenty-three days, until the mice in the PBS-control group reached humane endpoints (Figure 6A). The tumor volume was measured every 3–5 days, and mouse weights were monitored to assess potential toxicity. The tumors in mice receiving control treatments began growing exponentially around day 14, while the growth rate of tumors in mice that received the BL21-TAN and gemcitabine combination treatment was inhibited. By day 23, the tumor sizes in the BL21-TAN plus gemcitabine combination treatment group were significantly smaller (Figure 6B) than those treated with PBS (*p* = 0.0002, mixed-effects), gemcitabine only (*p* < 0.0001, mixed-effects), and BL21-TAN only (*p* < 0.0001, mixed-effects). These data indicate that the BL21-TAN pre-treatment enhanced the efficacy of gemcitabine in PDAC tumors. Furthermore, the tumor sizes were significantly lower in the BL21-TAN plus gemcitabine group compared to the BL21-eGFP plus gemcitabine group (*p* = 0.0006, mixed-effects), suggesting that the enhanced efficacy of the combination treatment is specifically due to the degradation of collagen and HA within the tumor by BL21-TAN. BL21-TAN monotherapy also appeared to decelerate tumor growth, albeit not as well as when in combination with chemotherapy. Desmoplasia has been linked to poor immune cell infiltration and is a negative predictor of immunotherapeutic response [52,53]. Therefore, we considered an increased immune infiltration, specifically macrophage infiltration, as a possible mechanism for the therapeutic effects seen with the BL21-TAN treatment. Tumor sections from mice treated with BL21-eGFP or BL21-TAN were stained for murine macrophage markers to determine the extent of infiltration, however, we saw no significant difference in the immune cell composition of these tumors (Appendix A). Mouse weights are commonly used as non-invasive assessments to determine systemic toxicity and sepsis due to bacterial infection in mammalian research [54,55]. We found that the mouse weights were comparable across all treatments, indicating that the treatment with BL21-TAN was not a source of systemic toxicity (Figure 6C). Taken together with our previous results, our study showed that the BL21-TAN pre-treatment depletes HA and collagen in PDAC tumors, leading to an increased efficacy of chemotherapeutic treatment.

## 4. Discussion

Desmoplasia is a hallmark of the PDAC which leads to obstructed drug penetration and immunosuppression, making effective therapeutic treatment difficult in PDAC patients [5,6]. Tumor metabolism, survival, and migration are promoted by abundant, aberrant collagen and HA in PDAC tumors [9,10,11,12,19,20]. For these reasons, the depletion of collagen and HA is anticipated to serve as an effective therapeutic intervention for desmoplastic tumors [36,56]. Various different avenues for targeting the fibrotic ECM are being explored, such as inducing the apoptosis of cells responsible for ECM deposition, inhibiting collagen synthesis and collagen crosslinking, and HA degradation to decrease interstitial fluid pressure. However, these therapies have had limited success in clinical trials due to difficulties with targeted delivery of molecules, insufficient tumor regression, and increased disease progression [57,58]. Therefore, the development of tumor-targeting ECM-depleting treatments is necessary.

Collagen and HA are integral components of the ECM present in most tissues. Therefore, to achieve targeted treatment while limiting systemic toxicity, we, along with other groups, previously employed tumor-colonizing bacteria engineered to deliver ECM-degrading enzymes directly to the tumor [30,59,60]. Here, we have engineered gram-negative bacteria to deliver both collagenase and hyaluronidase, as a single therapeutic agent, to PDAC tumor models. We show that the engineered bacteria successfully colonize murine models of PDAC tumors, express ECM-degrading enzymes, and deplete tumor-derived collagen and HA.

The degradation of ECM proteins increased biomaterial diffusion and distribution in tumor-like spheroids in vitro and intratumoral distribution in vivo [61,62,63,64]. Therefore, we theorized that the BL21-TAN pre-treatment would increase the penetration of gemcitabine, a small-molecule DNA synthesis inhibitor [65], in PDAC tumors following the degradation of the tumor ECM. We showed that tumor-bearing mice pre-treated with BL21-TAN followed by gemcitabine had increased survival and a decreased tumor burden compared to the mice treated with gemcitabine alone. This indicates that the BL21-TAN pre-treatment increases the efficacy of chemotherapy in PDAC models, most likely by creating a less fibrotic microenvironment which facilitates the diffusion of gemcitabine.

Tumor-derived collagen and HA differ in function and structure compared to healthy tissue and contribute to the immunosuppressive TME in solid tumors. Although we did not see an increased macrophage infiltration following BL21-TAN treatment, collagen and hyaluronan promote therapeutic resistance in solid tumors through signaling pathways and receptors, such as TGF-β, LAIR-1, and CD44 [7,66,67]. Leukocyte-associated immunoglobulin-like receptor-1 (LAIR-1), an immune inhibitory receptor, binds to collagen in the TME and leads to decreased immune activation and decreased immune cell infiltration [7,68]. HA promotes the differentiation of tumor-associated macrophages, resulting in increased immunosuppression within the TME through PD-1/PD-L1 signaling. Dense ECM also leads to interstitial fluid flow toward tumor margins, leading to the expression of TGF-β in cancer-associated fibroblasts, which polarizes regulatory T cells and immunosuppressive macrophage phenotypes [69,70]. Furthermore, increased organ stiffness, due to desmoplasia, is correlated to a poor immunotherapeutic response and has been suggested as a prognostic marker for immunotherapeutic intervention [53,71]. Therefore, future studies might consider looking into the activation status of various immune cells following treatment with BL21-TAN, as there is evidence that a fibrotic, tumor-associated ECM contributes to the immunosuppressive TME.

## 5. Conclusions

We genetically engineered *E. coli* strain BL21 to express both collagenase and hyaluronidase (BL21-TAN) within PDAC tumor models, and effectively deplete collagen and hyaluronan within the microenvironment. We show that pre-treatment with BL21-TAN in tumor-bearing mice leads to the increased efficacy of the chemotherapeutic agent, gemcitabine. This work, in addition to our previous work utilizing bacteria expressing only a single ECM-degrading enzyme, emphasizes the importance of eliminating desmoplasia in PDAC to maximize the effectiveness of therapy.

## Figures and Tables

**Figure 1 biomolecules-14-01458-f001:**
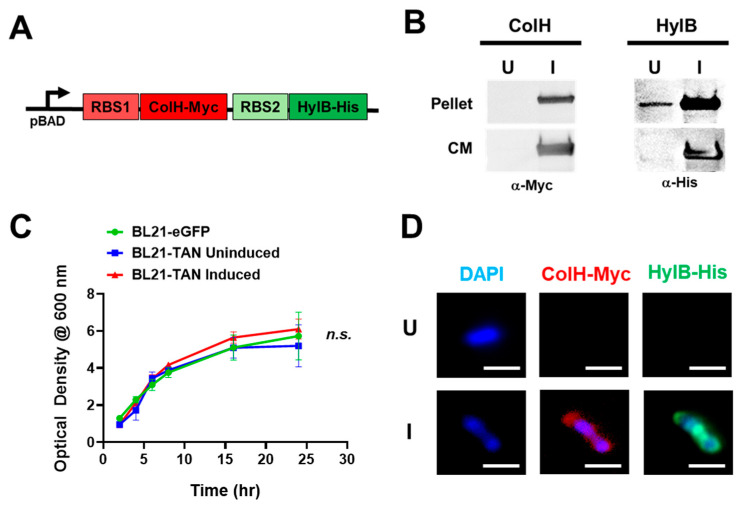
Tandem construct design and expression analysis in BL21 *E. coli* transformants. (**A**) The synthesized sequence encoding for both bacterial collagenase (ColH) and hyaluronidase (HylB) was engineered with independent 5′ ribosomal binding sites (RBSs) and fused to a myc- or his-tag, respectively. The sequence was cloned downstream of the inducible pBAD promoter in the pBAD/HIS A plasmid and then transformed into BL21 to generate BL21-TAN. (**B**) BL21-TAN cultures were grown to an exponential phase (shaking at 37 °C) and then left alone (uninduced, U) or induced (I) at a final concentration of 0.02% L-arabinose for 4 h. Bacterial pellets and culture media (CM) were then subjected to western blot analysis to detect the expression of ColH (anti-Myc) and HylB (anti-His). Western blot original images can be found in Appendix A. (**C**) BL21 transformed with a control pBAD-eGFP plasmid (BL21-eGFP), and BL21-TAN were cultured to an optical density (OD_600_) of ~1. Cultures were either left uninduced (BL21-eGFP, BL21-TAN) or induced at 0.02% L-arabinose (BL21-TAN), and OD_600_ was measured over time. (**D**) Uninduced or induced BL21-eGFP (0.02% L-arabinose, 4 h) were fixed in 4% paraformaldehyde and then stained simultaneously with anti-myc and anti-his to detect ColH and HylB expression, respectively. A representative single bacterium for each condition is shown. *n.s*. = no significance. Error bars display standard error of the mean.

**Figure 2 biomolecules-14-01458-f002:**
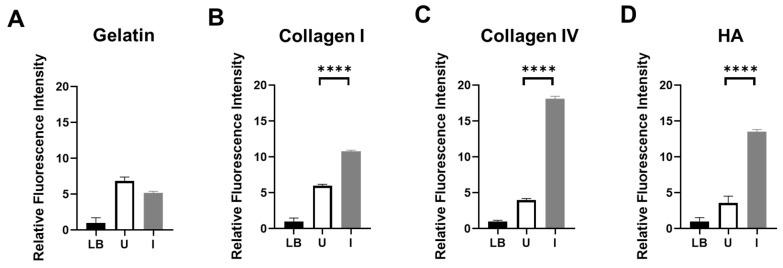
In vitro degradation of collagens and hyaluronic acid by BL21-TAN. Hydrolysis reactions were performed using uninduced (U) or induced (I) BL21-TAN co-incubated with FITC-conjugated pig skin gelatin (**A**), bovine skin collagen type I (**B**), human placenta collagen type IV (**C**), or purified hyaluronic acid (HA) (**D**) in 50 mM Tris-HCl (pH 8.0) containing 10 mM CaCl_2_ at 37 °C. The negative control includes the culture media (LB). Increases in fluorescence intensity signify the degradation of the FITC-conjugated target. Enzyme activity was measured by monitoring fluorescence (FITC) (ex: 495 nm, em: 519 nm). Data are expressed as mean ± error of the mean of three independent experiments. **** *p* < 0.0001, *t*-test.

**Figure 3 biomolecules-14-01458-f003:**
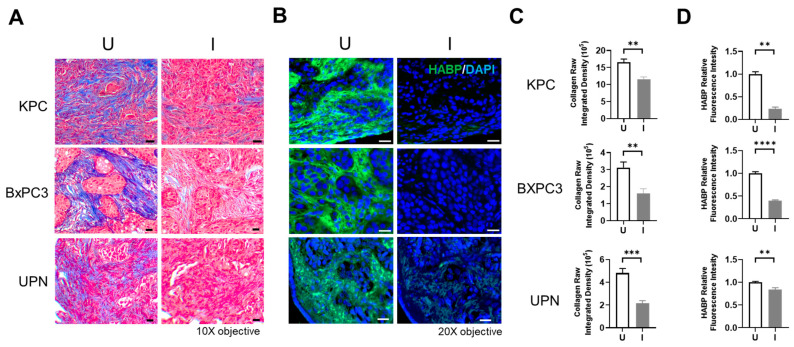
BL21-TAN depletes PDAC-derived collagen and HA in serial tumor sections. Serial sections of KPC, BxPC3, and de-identified patient (UPN) PDAC tumors were treated overnight with BL21-TAN under uninduced or induced conditions at 37 °C. Sections were then stained by trichrome to detect collagen (blue). (**A**) or biotin-labeled hyaluronic acid binding protein (HABP) followed by streptavidin-FITC (**B**). Trichrome images were deconvoluted using ImageJ to quantify collagen content (blue staining) in randomly selected fields (10) of each tumor section (**C**). Fluorescence intensity was used to quantify HA content (FITC/488 channel) and was measured using ImageJ and normalized to uninduced treatment (**D**). Data are expressed as mean values ± error of the mean. ** *p* < 0.01; *** *p* < 0.001; **** *p* < 0.0001, *t*-test. Scale bar = 20 µm.

**Figure 4 biomolecules-14-01458-f004:**
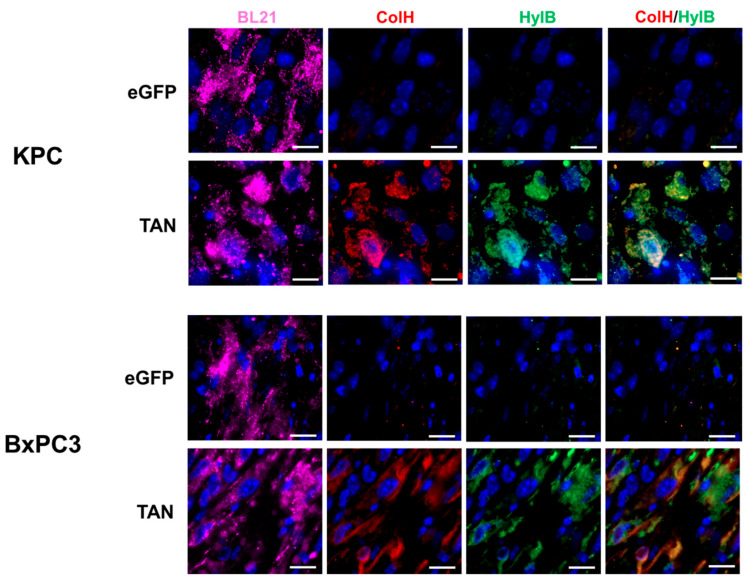
Intravenously administered BL21-TAN colonizes PDAC tumors and expresses both ColH and HylB. Mice bearing subcutaneous KPC or BxPC3 tumors (6–8 mm diameter) were intravenously injected with 5 × 10^7^ colony-forming units (CFUs) of BL21-TAN for three consecutive days. Twenty-four hours following the final injection, the mice were either administered PBS (uninduced, U) or 40 mg L-arabinose (induced, I) intraperitoneally. Tumors were collected 48 h post induction and sections were evaluated for BL21-TAN colonization and enzyme expression by immunofluorescence using antibodies specific to BL21 *E. coli*, Myc-tag (ColH) and His-tag (HylB). Objective: 100× oil. Scale bars = 10 µm.

**Figure 5 biomolecules-14-01458-f005:**
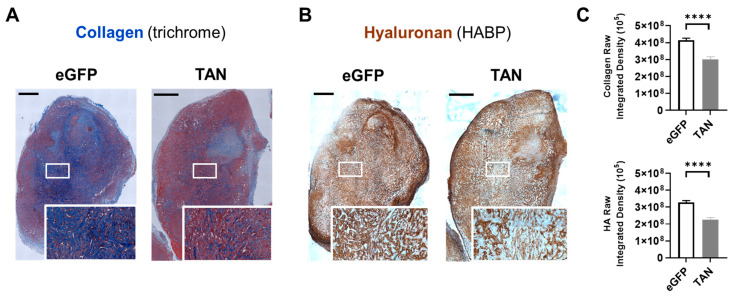
In vivo collagen and HA depletion by BL21-TAN. NSG mice with subcutaneous BxPC3 xenografts (6–8 mm diameter) were intravenously injected with 5 × 10^7^ colony-forming units (CFUs) of the BL21-eGFP control or BL21-TAN for three consecutive days by the intravenous route. Twenty-four hours following the final injection, the mice given BL21-TAN were administered 40 mg L-arabinose intraperitoneally. Tumors (*n* = 8) were collected 48 h post induction and serial sections were evaluated for collagen (blue) using trichrome staining (**A**) and HA using HABP staining (**B**). Regions (box) from each image were magnified for greater resolution of collagen and HA (inset). Random fields (*n* = 15, 10× objective) from each treatment group were used for deconvolution analysis to quantify collagen and HA content in multiple tumors (**C**). Data are expressed as mean ± error of the mean. **** *p* < 0.0001, *t*-test. Scale bars = 1 mm.

**Figure 6 biomolecules-14-01458-f006:**
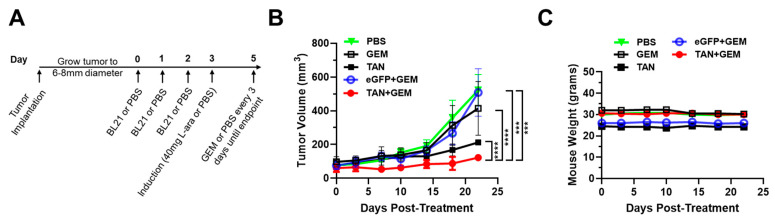
BL21-TAN pre-treatment of PDAC tumors enhances the efficacy of gemcitabine. (**A**) NSG mice with subcutaneous BxPC3 xenografts (6–8 mm diameter) were injected with 5 × 10^7^ colony-forming units (CFUs) of the BL21-eGFP control or BL21-TAN for three consecutive days by the intravenous route. After 24 h of the final injection, mice were administered 40 mg L-arabinose (induction) or PBS control intraperitoneally. Gemcitabine (GEM) or vehicle (PBS) was injected intraperitoneally 48 h after induction and maintenance doses were given every three days thereafter. Tumor growth (**B**) and mouse weights (**C**) were measured over time until control groups required euthanization. Data are expressed as mean values ± error of the mean. *** *p* < 0.001; **** *p* < 0.0001, 2-way ANOVA.

## Data Availability

Data are contained within the article and Appendix A.

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
