# Peer review of "Tumor-Colonizing E. coli Expressing Both Collagenase and Hyaluronidase Enhances Therapeutic Efficacy of Gemcitabine in Pancreatic Cancer Models"

_biomolecules, 2024, doi:10.3390/biom14111458_

Round 1
Reviewer 1 Report
Comments and Suggestions for Authors
Avsharian et al. have explored the therapeutic potential of using an engineered tumor-colonizing, E.coli-based agent that expresses both collagenase and hyaluronidase as a strategy to reduce desmoplasia and enhance intratumoral perfusion of anticancer agents in PDAC. The manuscript addresses an on-going important issue in the treatment of PDAC and is well-written. However, there are a few points from which the manuscript could be improved:
1. Do the KPC cell line and BxPC3 cell line tumours used in Fig3 truly recapitulate the histological hallmarks of human PDAC, in particular the desmoplasia?
2. The images of the Trichrome staining of the KPC, BxPC3 and UPN tumours in Fig3A do not look significantly reduced in the induced sections, in fact they seem to have greater expression than the uninduced which is contrary to the results in the graph. Also it would have been interesting to display the results of Collagen and HABP expression based on tumour expression and stroma expression?
3. In figure 4 was the BL21 staining observed specifically in tumour cells and/or in the surrounding stroma? A chromogenic stain at lower magnification or dual staining with a tumour specific marker would definitively show this.
4. In Figure 4 the staining expression pattern of Co1H and Hy1B looks exactly the same? Were the Abs used to detect these directly conjugated Abs and how did you control for any spectral bleed-through of the fluorescent channels?
5. As in Fig 3, in Fig5A the collagen looks significantly more in the TAN than eGFP treated tumours? Again in both Fig5A and B it would have been interesting to define the expression of collagen and HABP based on tumour expression vs stromal expression.
Were the tumours from the in-vivo exp in Fig6 harvested and analysis of tumour ECM performed? Did the combined treatment affect the presence/phenotype of the immune infiltrate? The authors themselves stated in their discussion that the infiltration and activation status of immune cells following treatment with BL21-TAN should be studied so it was disappointing that the authors did not take the opportunity to look at this as part of their current study? This would have definitively shown the underlying mechanism behind the enhanced treatment efficacy of the BL21-TAN agent, providing important information for future therapeutic strategies in PDAC.
Author Response
- “Do the KPC cell line and BxPC3 cell line tumours used in Fig3 truly recapitulate the histological hallmarks of human PDAC, in particular the desmoplasia?”
- This is a very valid point. As with any immortalized cell line, no, they do not completely recapitulate the histological hallmarks of human PDAC. However, they do recapitulate certain aspects. For example, increased interstitial fluid pressure, hypoxia, and hypovascularity are seen in KPC models which are also seen in human tumor tissue [1-4]. BxPC3 cells are commonly used in both in vitro and in vivo experimental models of PDAC stroma, with excess deposition of type I collagen (one of the most abundant collagens in human PDAC) and extensive collagen remodeling, leading to increased stiffness (a prognostic marker for human PDAC therapeutic response), in BxPC3 models [5, 6]. However, in efforts to recapitulate the histological hallmarks of human PDAC more closely, we include the UPN model in Figure 3, last row, which was derived from a de-identified PDAC patient tumor biopsy, here at the City of Hope.
- To best address Reviewer 1 concerns, we have now included a description of the models with additional references, justifying their use in our studies, on page 6 of the manuscript, line 240-244.
- “The images of the Trichrome staining of the KPC, BxPC3 and UPN tumours in Fig3A do not look significantly reduced in the induced sections, in fact they seem to have greater expression than the uninduced which is contrary to the results in the graph. Also it would have been interesting to display the results of Collagen and HABP expression based on tumour expression and stroma expression?”
- Respectfully, we disagree with this assessment. Trichrome staining allows for the analysis of collagen deposition with the application of aniline blue stain. With trichrome staining, nuclei appear dark red or purple, the cytoplasm appears pink, and collagen appears blue [7]. In Fig3A, we see the presence of blue color in tumor sections of all three PDAC models (rows) in the uninduced group (left column) and less blue color in the induced group (right column), indicating less collagen density within the induced group (the group expressing collagenase).
- We have now edited the manuscript to clarify this on page 7, figure legend 3, and page 6 line 247.
- From previous studies, we know that pancreatic stellate cells and cancer-associated fibroblasts are the primary cells that deposit collagen in PDAC tumors [8]. Hyaluronic acid in PDAC tumors is secreted from both tumor cells and stromal cells [9].
- We have now edited the manuscript to include the results from these previous studies, with additional references, on page 2, line 56-58 and 61-62.
- “In figure 4 was the BL21 staining observed specifically in tumour cells and/or in the surrounding stroma? A chromogenic stain at lower magnification or dual staining with a tumour specific marker would definitively show this.”
- Our objective with Fig4 was to determine whether we had BL21 colonization and expression of both enzymes within tumor tissue in vivo following systemic administration, which is why we chose to image this way. BL21, however, is a non-pathogenic laboratory strain of coli that lacks adhesion and invasion genes necessary for invasion into human cells. It may enter human cells if internalized by the host cell through engulfment. (PMID: 11123478) Furthermore, Zare et al. previously showed that native BL21 does not invade human cell lines. (PMID: 37166604) Therefore, we would expect that the majority of the BL21 remains extracellular.
- “In Figure 4 the staining expression pattern of Co1H and Hy1B looks exactly the same? Were the Abs used to detect these directly conjugated Abs and how did you control for any spectral bleed-through of the fluorescent channels?”
- In the overlay images in the last column of the second and fourth row of Figure 4, we see that there is a similar expression pattern of ColH and HylB in the tissue (orange/yellow staining), which is expected as our construct is designed to allow ColH and HylB expression from a single bacterium, as we show in Figure 1D. However, we also see green (HylB) staining alone and red staining (ColH) alone in these panels, suggesting that there is no spectral bleed-through, and potentially the presence of bacteria that express one enzyme or the other. To further clarify, these were not directly conjugated antibodies and secondary antibodies were used.
- The reviewer does make a good point, however, and we have further discussed the staining in this figure in the manuscript on page 7, line 279-282.
- “As in Fig 3, in Fig5A the collagen looks significantly more in the TAN than eGFP treated tumours? Again in both Fig5A and B it would have been interesting to define the expression of collagen and HABP based on tumour expression vs stromal expression.”
- We again respectfully disagree with this assessment. As in Figure 3, trichrome staining allows for collagen visualization in the blue color, and we see that there is less blue overall in the tumor section on the right in figure 5A, indicating less collagen density. With this figure, our goal was to determine the functional capacity of our BL21-TAN treatment Our treatment specifically targets collagen and hyaluronan within the tumor, therefore we decided to quantify the depletion of these molecules. The treatment does not target cells expressing collagen and HABP, therefore we would not expect collagen and HABP expression to differ, whether it be in tumor cells or stromal cells.
- We have clarified this assessment in the manuscript on page 9, figure legend 5 and page 8, line 290.
- “Were the tumours from the in-vivo exp in Fig6 harvested and analysis of tumour ECM performed?”
- The tumors from Fig6 were not harvested and assessed. However, due to our in vivo tumor analysis of ECM in Fig5, we see that BL21-TAN does indeed reduce collagen and hyaluronan in the tumor, therefore we would expect the same within the tumors in Fig6. From previous studies, we know the mechanism of gemcitabine is inhibition of DNA synthesis and would not expect it to have an effect on ECM degradation.
- We now reference these studies in the text on page 10, line 350-357.
- “Did the combined treatment affect the presence/phenotype of the immune infiltrate?”
- The combined treatment was performed in NSG mice, and at initial assessment of the innate immune cells, specifically F4/80+ macrophages, we did not see an increase in the frequency of these cells. However, collagen has been shown to bind LAIR-1, an inhibitory receptor highly expressed on macrophages that influences their phenotype. LAIR-1 activation by collagen inhibits the M1-like polarization of macrophages and also inhibits the differentiation of monocytes in the periphery [10-12]. Therefore, we hypothesize that with BL21-TAN treatment and subsequent collagen depletion within the tumor, we will see a change in the phenotype of the immune infiltration.
- We have included this observation as new data in the manuscript in a supplemental figure referenced on page 9, line 327-333 and page 10-11, line 369-372 with additional references.
- “The authors themselves stated in their discussion that the infiltration and activation status of immune cells following treatment with BL21-TAN should be studied so it was disappointing that the authors did not take the opportunity to look at this as part of their current study? This would have definitely shown the underlying mechanism behind the enhanced treatment efficacy of the BL21-TAN agent, providing important information for future therapeutic strategies in PDAC.”
- We did not further investigate the activation status of the immune cells, due to the murine model we had used in this study and our results on macrophage infiltration into the tumor, which we have now included in the manuscript. We believe that the underlying mechanism behind the enhanced therapeutic efficacy is due to increased drug penetration. Although we do not have an assay to detect gemcitabine diffusion within the tissue, previous studies have assessed the change in small-molecule distribution following degradation of ECM molecules, showing that upon ECM degradation, small-molecules are better able disseminate further into the tumor tissue and affect more tumor cells [4].
- We have reflected on the results in the above-mentioned studies further with new references in our manuscript on page 10, line 356-358.
- Whatcott CJ, Diep CH, Jiang P, Watanabe A, LoBello J, Sima C, Hostetter G, Shepard HM, Von Hoff DD, Han H. Desmoplasia in Primary Tumors and Metastatic Lesions of Pancreatic Cancer. Clin Cancer Res. 2015;21(15):3561-8. Epub 2015/02/20. doi: 10.1158/1078-0432.CCR-14-1051. PubMed PMID: 25695692; PMCID: PMC4526394.
- Olive KP, Jacobetz MA, Davidson CJ, Gopinathan A, McIntyre D, Honess D, Madhu B, Goldgraben MA, Caldwell ME, Allard D, Frese KK, Denicola G, Feig C, Combs C, Winter SP, Ireland-Zecchini H, Reichelt S, Howat WJ, Chang A, Dhara M, Wang L, Ruckert F, Grutzmann R, Pilarsky C, Izeradjene K, Hingorani SR, Huang P, Davies SE, Plunkett W, Egorin M, Hruban RH, Whitebread N, McGovern K, Adams J, Iacobuzio-Donahue C, Griffiths J, Tuveson DA. Inhibition of Hedgehog signaling enhances delivery of chemotherapy in a mouse model of pancreatic cancer. Science. 2009;324(5933):1457-61. Epub 2009/05/23. doi: 10.1126/science.1171362. PubMed PMID: 19460966; PMCID: PMC2998180.
- Cannon A, Thompson C, Hall BR, Jain M, Kumar S, Batra SK. Desmoplasia in pancreatic ductal adenocarcinoma: insight into pathological function and therapeutic potential. Genes Cancer. 2018;9(3-4):78-86. Epub 2018/08/16. doi: 10.18632/genesandcancer.171. PubMed PMID: 30108679; PMCID: PMC6086006.
- Jacobetz MA, Chan DS, Neesse A, Bapiro TE, Cook N, Frese KK, Feig C, Nakagawa T, Caldwell ME, Zecchini HI, Lolkema MP, Jiang P, Kultti A, Thompson CB, Maneval DC, Jodrell DI, Frost GI, Shepard HM, Skepper JN, Tuveson DA. Hyaluronan impairs vascular function and drug delivery in a mouse model of pancreatic cancer. Gut. 2013;62(1):112-20. Epub 2012/04/03. doi: 10.1136/gutjnl-2012-302529. PubMed PMID: 22466618; PMCID: PMC3551211.
- Suklabaidya S, Dash P, Das B, Suresh V, Sasmal PK, Senapati S. Experimental models of pancreatic cancer desmoplasia. Lab Invest. 2018;98(1):27-40. Epub 2017/11/21. doi: 10.1038/labinvest.2017.127. PubMed PMID: 29155423.
- Kim SK, Jang SD, Kim H, Chung S, Park JK, Kuh HJ. Phenotypic Heterogeneity and Plasticity of Cancer Cell Migration in a Pancreatic Tumor Three-Dimensional Culture Model. Cancers (Basel). 2020;12(5). Epub 2020/05/28. doi: 10.3390/cancers12051305. PubMed PMID: 32455681; PMCID: PMC7281339.
- Van De Vlekkert D, Machado E, d'Azzo A. Analysis of Generalized Fibrosis in Mouse Tissue Sections with Masson's Trichrome Staining. Bio Protoc. 2020;10(10):e3629. Epub 2021/03/05. doi: 10.21769/BioProtoc.3629. PubMed PMID: 33659302; PMCID: PMC7842772.
- Sahai E, Astsaturov I, Cukierman E, DeNardo DG, Egeblad M, Evans RM, Fearon D, Greten FR, Hingorani SR, Hunter T, Hynes RO, Jain RK, Janowitz T, Jorgensen C, Kimmelman AC, Kolonin MG, Maki RG, Powers RS, Pure E, Ramirez DC, Scherz-Shouval R, Sherman MH, Stewart S, Tlsty TD, Tuveson DA, Watt FM, Weaver V, Weeraratna AT, Werb Z. A framework for advancing our understanding of cancer-associated fibroblasts. Nat Rev Cancer. 2020;20(3):174-86. Epub 2020/01/26. doi: 10.1038/s41568-019-0238-1. PubMed PMID: 31980749; PMCID: PMC7046529 Cancer Research Center receives research funding. R.K.J. received honorarium from Amgen, consultant fees from Chugai, Merck, Ophthotech, Pfizer, SPARC, SynDevRx and XTuit, owns equity in Enlight, Ophthotech and SynDevRx and serves on the boards of trustees of Tekla Healthcare Investors, Tekla Life Sciences Investors, Tekla Healthcare Opportunities Fund and Tekla World Healthcare Fund. Neither any reagent nor any funding from these organizations was used in this study. A.C.K. has financial interests in Vescor Therapeutics, is an inventor named on patents pertaining to KRAS-regulated metabolic pathways, redox control pathways in pancreatic cancer, targeting GOT1 as a therapeutic approach and the autophagic control of iron metabolism, is on the Scientific Advisory Board of Rafael Pharmaceuticals and has been a consultant for Deciphera Pharmaceuticals. R.G.M. receives consulting fees from Bayer, Deciphera, Karyopharm, Springworks, the American Society of Clinical Oncology and UptoDate. E.P. receives research funds in the form of a sponsored research agreement from TMUNITY and a collaborative research agreement from Boehringer Ingelheim; prior funding was provided by Novartis. D.A.T. has received commercial research grants from Fibrogen and ONO, has ownership interest (including stock, patents and so on) in Leap Therapeutics and Surface Oncology and is a consultant/advisory board member for Leap Oncology, Surface Oncology, Cygnal and Merck. D.A.T. is Director and Chief Scientist of the Lustgarten Foundation, a designated laboratory of pancreatic cancer research. F.M.W. is currently on secondment as Executive Chair of the UK Medical Research Council. Z.W. is on the Advisory Board of Maverick Therapeutics. E.S., I.A., E.C., D.D., M.E., R.M.E., D.F., F.R.G., T.H., R.O.H., T.J., C.J., M.G.K, R.S.P., D.C.R., R.S-S., M.H.S., S.S., T.D.T., V.W. and A.T.W. declare no competing interests.
- Sato N, Kohi S, Hirata K, Goggins M. Role of hyaluronan in pancreatic cancer biology and therapy: Once again in the spotlight. Cancer Sci. 2016;107(5):569-75. Epub 2016/02/27. doi: 10.1111/cas.12913. PubMed PMID: 26918382; PMCID: PMC4970823.
- Jin J, Wang Y, Ma Q, Wang N, Guo W, Jin B, Fang L, Chen L. LAIR-1 activation inhibits inflammatory macrophage phenotype in vitro. Cell Immunol. 2018;331:78-84. Epub 2018/06/12. doi: 10.1016/j.cellimm.2018.05.011. PubMed PMID: 29887420.
- Poggi A, Tomasello E, Ferrero E, Zocchi MR, Moretta L. p40/LAIR-1 regulates the differentiation of peripheral blood precursors to dendritic cells induced by granulocyte-monocyte colony-stimulating factor. Eur J Immunol. 1998;28(7):2086-91. Epub 1998/08/06. doi: 10.1002/(SICI)1521-4141(199807)28:07<2086::AID-IMMU2086>3.0.CO;2-T. PubMed PMID: 9692876.
- Van Laethem F, Donaty L, Tchernonog E, Lacheretz-Szablewski V, Russello J, Buthiau D, Almeras M, Moreaux J, Bret C. LAIR1, an ITIM-Containing Receptor Involved in Immune Disorders and in Hematological Neoplasms. Int J Mol Sci. 2022;23(24). Epub 2022/12/24. doi: 10.3390/ijms232416136. PubMed PMID: 36555775; PMCID: PMC9788452.
Reviewer 2 Report
Comments and Suggestions for Authors
Manuscript entitled “Tumor-Colonizing E. coli Expressing both Collagenase and Hyaluronidase Enhances Therapeutic Efficacy of Gemcitabine in Pancreatic Cancer Models” by Avsharian et al. 2024 submitted to “biomolecules” has been reviewed. Reviewer does not agree with its publication at this stage. Some revisions are needed for improvement of paper. To improve the quality authors should revise these points.
Author shoud clarify the expression “Furthermore, mouse weights were comparable across all treatments, indicating that treatment with BL21-TAN was not a source of increased toxicity (Figure 6C)” in the text. How authors decided to BL21-TAN was not a source of increased toxicity? Exclussion of toxicity should be detailed.
Conclusion paragraph should ultimately suggest to your reader that you've accomplished what you set out to prove but “Conclusions” section in manuscript is too short. This section should be re-written.
Author Response
“Manuscript entitled “Tumor-Colonizing E. coli Expressing both Collagenase and Hyaluronidase Enhances Therapeutic Efficacy of Gemcitabine in Pancreatic Cancer Models” by Avsharian et al. 2024 submitted to “biomolecules” has been reviewed. Reviewer does not agree with its publication at this stage. Some revisions are needed for improvement of paper. To improve the quality authors should revise these points.
Author should clarify the expression “Furthermore, mouse weights were comparable across all treatments, indicating that treatment with BL21-TAN was not a source of increased toxicity (Figure 6C)” in the text. How authors decided to BL21-TAN was not a source of increased toxicity? Exclusion of toxicity should be detailed.”
- Body weight is a commonly used, non-invasive monitoring tool to observe the systemic toxicity of an experimental treatment in mammals, with 5% loss in body weight to be associated with a strong predictor of systemic toxicity, a 10% or more loss to be concerning and a 20% decrease in normal body weight to be means for humane euthanization [1, 2].
- We have clarified this with additional text and references on page 9, line 324-326.
“Conclusion paragraph should ultimately suggest to your reader that you've accomplished what you set out to prove but “Conclusions” section in manuscript is too short. This section should be re-written.”
- We thank the reviewer for their insight. We have re-written this section and expanded on the conclusions of the manuscript on page 11, line 376-379.
- Brochut M, Heinonen T, Snaka T, Gilbert C, Le Roy D, Roger T. Using weight loss to predict outcome and define a humane endpoint in preclinical sepsis studies. Sci Rep. 2024;14(1):21150. Epub 2024/09/11. doi: 10.1038/s41598-024-72039-1. PubMed PMID: 39256525; PMCID: PMC11387420.
- Silva AV, Norinder U, Liiv E, Platzack B, Oberg M, Tornqvist E. Associations between clinical signs and pathological findings in toxicity testing. ALTEX. 2021;38(2):198-214. Epub 2020/10/30. doi: 10.14573/altex.2003311. PubMed PMID: 33118607.
Reviewer 3 Report
Comments and Suggestions for Authors
The manuscript by Avsharian et al., “Tumor-Colonizing E. coli Expressing both Collagenase and Hyaluronidase Enhances Therapeutic Efficacy of Gemcitabine in Pancreatic Cancer Models” seems to be interesting, well written and the data are clear. The methods are well described.
However, there is some open question about the effect of the depletion of both collagen and hyaluronan on the tissues of other organs of the mice? Such as skin and other internal organs
Author Response
“The manuscript by Avsharian et al., “Tumor-Colonizing E. coli Expressing both Collagenase and Hyaluronidase Enhances Therapeutic Efficacy of Gemcitabine in Pancreatic Cancer Models” seems to be interesting, well written and the data are clear. The methods are well described.
However, there is some open question about the effect of the depletion of both collagen and hyaluronan on the tissues of other organs of the mice? Such as skin and other internal organs.”
- We thank the reviewer for their laudatory assessment. In response to their concern, in BL21 colonization studies, 3 days post-treatment, we did not detect BL21 in the spleen, indicating that it had cleared from systemic circulation. In previous studies, researchers have assessed the biosafety of BL21 as a therapeutic agent, showing that BL21 preferentially colonizes tumor tissue, with limited detection in systemic tissues such as the heart, liver, spleen, lung and kidney, along with effective clearance of the bacteria over time by the host immune system [1]. In our previously published work that evaluated a collagenase-only expressing bacterial vector, we did address this question directly by assessing the extent of collagen depletion within the skin and joints, locations with high collagen content, and did not detect a reduction in collagen, nor did we see bacterial colonization [2].
- We have now addressed this concern in the manuscript with new text/references on page 2, line 82-83.
- Yang H, Jiang F, Ji X, Wang L, Wang Y, Zhang L, Tang Y, Wang D, Luo Y, Li N, Wang Q, Zou J. Genetically Engineered Bacterial Protein Nanoparticles for Targeted Cancer Therapy. Int J Nanomedicine. 2021;16:105-17. Epub 2021/01/16. doi: 10.2147/IJN.S292432. PubMed PMID: 33447030; PMCID: PMC7802776.
- Ebelt ND, Zamloot V, Zuniga E, Passi KB, Sobocinski LJ, Young CA, Blazar BR, Manuel ER. Collagenase-Expressing Salmonella Targets Major Collagens in Pancreatic Cancer Leading to Reductions in Immunosuppressive Subsets and Tumor Growth. Cancers (Basel). 2021;13(14). Epub 2021/07/25. doi: 10.3390/cancers13143565. PubMed PMID: 34298778; PMCID: PMC8306875.
Round 2
Reviewer 2 Report
Comments and Suggestions for Authors
accept as is